# A Visual Enhancement Network with Feature Fusion for Image Aesthetic Assessment

**Xin Zhang [1], Xinyu Jiang [2,\*], Qing Song [3] and Pengzhou Zhang [4]**

1   School of Computer and Cyber Sciences, Communication University of China, Beijing 100024, China; rebeccazhang@cuc.edu.cn
2   Institute of Information and Communication Engineering, Communication University of China, Beijing 100024, China
3   Convergence Media Center, Communication University of China, Beijing 100024, China; songqing@cuc.edu.cn
4   State Key Laboratory of Media Convergence and Communication, Communication University of China, Beijing 100024, China; zhangpengzhou@cuc.edu.cn
\*   Correspondence: jiangxinyu@cuc.edu.cn; Tel.: +86-(10)-6577-9210

**Abstract:** Image aesthetic assessment (IAA) with neural attention has made significant progress due to its effectiveness in object recognition. Current studies have shown that the features learned by convolutional neural networks (CNN) at different learning stages indicate meaningful information. The shallow feature contains the low-level information of images, and the deep feature perceives the image semantics and themes. Inspired by this, we propose a visual enhancement network with feature fusion (FF-VEN). It consists of two sub-modules, the visual enhancement module (VE module) and the shallow and deep feature fusion module (SDFF module). The former uses an adaptive filter in the spatial domain to simulate human eyes according to the region of interest (ROI) extracted by neural feedback. The latter not only extracts the shallow feature and the deep feature via transverse connection, but also uses a feature fusion unit (FFU) to fuse the pooled features together with the aim of information contribution maximization. Experiments on standard AVA dataset and Photo.net dataset show the effectiveness of FF-VEN.

**Keywords:** deep learning; image aesthetics assessment; image enhancement

## 1. Introduction

Image aesthetic assessment (IAA) and image quality assessment (IQA) are popular image evaluation methods from two different directions. Usually, IAA focuses on graphic aesthetics and IQA focuses on the degree of distortion in a series of images. With the increasing application of digital images, the studies of IAA have made significant development. IAA has favorable commercial application value and potential. Images with high scores are considered better, which can be used in application scenarios such as recommendations and searches. Undeniably, several works on IQA achieve great results, including GraphIQA [1] and metaIQA [2]. Feature extraction has long been a question of great interest in IAA [3–5]. Early studies focused on photographic methods and human perception. Deng et al. [4] summarized the manual production features and the deep features, indicating the limitations of machine learning. Among the early efforts, the diversity of aesthetic features and the complexity of photographic methods resulted in the poor performance of the models with manual extraction.

Recently, deep learning has become a hot topic for IAA [5], which overcomes the limitation of hand-crafted feature extraction. Neural networks have shown good advantages for image analysis and processing [6–8]. Dai et al. [9] introduced the existing research in the field of intelligent media. Pooling layers were utilized to increase the speed of processing the low-level features [10]. Talebi et al. [11] modified the last layer of convolutional neural networks (CNN), directly predicting the aesthetic score distribution. This transforms the

classification model into a distributed task, increasing the training speed and improving the performance of neural networks. In [12], VGGNet stacks the convolutional layers, pushing the network depth to more than 16 weight layers. Ref. [13] showed that the intermediate convolution layers of CNN contain meaningful information about the complexity of images. Thus, we analyze the network structure and try to fuse the features of CNN, aiming to combine the low-level information and the abstract image semantics.

Neural attention is a major area of interest within the field of IAA. The question of how to assess digital images based on human visual characteristics is challenging for researchers. The deep model (TDM) was designed to perceive image scenes with the advantages of peripheral vision and central vision [14]. Ma et al. [15] introduced A-Lamp, which can learn the detailed and overall information of images. The details of images are retained via dynamically selecting image blocks. The overall information is extracted from the attribute graphs of the image blocks. Zhang et al. [16] combined the spatial layout and the details of images on the basis of top-down neural feedback. Inspired by TDM [14], they also proposed GLFN-Net [17]. It calculates the image blocks of region of interest (ROI) and simulates the fovea vision. However, human observation of images is flexible, not in the fixed shape of a rectangle. We attempt to dynamically process the digital images according to ROI.

In this paper, we propose a visual enhancement network with feature fusion (FF-VEN). It is divided into two sub-modules: the visual enhancement module (VE module) and the shallow and the deep feature fusion module (SDFF module). The VE module simulates the human eyes based on the fovea visual characteristics. It adaptively filters the images according to ROI obtained by neural feedback. The SDFF module consists of feature extraction and feature fusion. The shallow feature and the deep feature are extracted via the method of transverse connection. We design a feature fusion unit (FFU) that performs weighted fusion to maximize the contribution to information. The aesthetic score distribution is learned by minimizing squared earth mover's distance (EMD) loss. Further, FF-VEN is evaluated in the classification task and the regression task. Therefore, the contributions made by this paper are as follows:

(1) We propose an end-to-end training network, consisting of two sub-modules. The former module considers top-down neural attention and fovea visual characteristics. The latter module extracts and integrates the features learned by CNN at different stages.

(2) An adaptive filter is designed to select the filters in the spatial domain. Specifically, each pixel in the images adjusts the parameters of filters according to the normalized interest matrix extracted by neural feedback.

(3) We optimize a feature fusion unit to combine the low-level information and image semantics. The added pooling layers deal with the corresponding features, increasing the training speed and improving the precision of the predicted score prediction. Moreover, it fuses the features for contribution maximization.

The rest of the article is structured as follows. Section 2 introduces the relevant work briefly and Section 3 describes the proposed FF-VEN. Section 4 evaluates the performance of the network and compares it with other models. In Section 5, we summarize the paper.

## 2. Related Work

There are two basic approaches currently being adopted in research into IAA. One is extracting the image features manually and the other is based on deep learning. On the one hand, hand-crafted feature extraction means designing the aesthetic attributes of digital images on a computer based on photography, psychology, aesthetics, and other subjects. Datta et al. [18] defined the aesthetic image features, including color, structure, and image content, aiming to explore the relationship between human emotions and the low-level features. Reference [19] depended on the global saliency map of images and located the region of visual attention. From a photographer's point of view, Dhar et al. [20] analyzed image layout, sky lighting, and other image attributes. The relative foreground position and visual weight ratio are combined to enhance the visual image features [21]. Tang

et al. [22] integrated the regional and global features according to the eye-catching areas. They used a support vector machine (SVM) model for the classification task. However, the methods of hand-crafted feature extraction are unsuitable for all images. It creates a bottleneck, because aesthetics are abstract and the photographic methods are diverse.

On the other hand, deep learning has significant advantages for IAA [4]. For multi-scale image processing, Szegedy et al. [23] proposed GoogLeNet, increasing the width of the network via sparse connections. Because of its great performance on ImageNet, they developed InceptionNet, using optimization algorithms to improve the performance of the model [9]. DMA-NET [24] performed random image clipping and extracted local fine-grained information. A-Lamp learned the detailed information and the overall attributes of the input images [15]. Based on GoogLeNet [23], Jin et al. [25] considered the local and global views of images. They proposed ILG-NET, combing the InceptionNet and the connected layer. Yan et al. [26] obtained the aesthetic image features, including semantics, texture, and color. They weighted points of interest (POI) and segmented the image pixels. They proposed a circular attention network, which ignores irrelevant information and focuses on the attention region when extracting visual features [27]. From the gray value, contrast and spatial position relationships of pixels in color channels, the shallow feature perceives the image attributes, such as light, tone, clarity, and composition. The semantic information contains image object, theme, context, etc.

At present, the multi-channel frameworks have been widely used for IAA. She et al. [28] captured the image layout, using a special neural network composed of two sub-networks. A pooling layer of the multimodal factorized bilinear (MFB) was used to combine the features [27]. Based on [28], the GIF module integrated the weight generator into the feature fusion part [17]. They down-sampled the images to simulate peripheral vision, which missed the details and failed to assess the high-variance images. A gating unit (GU) performs dynamic weighted combination [29]. GU adds two fully connected layers and a Tanh layer, improving the effectiveness of the networks. It calculates the contribution of features to the result via analyzing the statistical characteristics. Inspired by this, we propose FFU, adding pooling layers for corresponding features based on GU. Ma et al. [15] showed that the ROI captured the spatial layout information of images and calculated the attention area of CNN. Zhang et al. [16] simulated fovea vision by generating image blocks via top-down neural feedback. Similarly, we use the incentive support method [30] to extract the interest matrix of images. However, the area of visual interest is not in the shape of a rectangle when humans assess images. We develop an adaptive filter, which is a pixel-based approach that captures the fine-grained details of an image in any shape.

Early studies treated IAA as a task of aesthetic classification [15,31]. According to the aesthetic score distribution, the average of scores is compared with the threshold value, aiming to divide the images into the high quality images and the low quality images. The aesthetic score distribution is ordered in IAA. Cross-entropy loss in the classification ignores the relationships between scores. The regression model was utilized to assess images [4]. For ordered classes, Zhang et al. [32] showed that models with the classification task can outperform regression networks. Due to the cultural background, the emotion states, the physiological condition, and other factors of the assessors, the aesthetic scores are highly subjective. At present, the research mainly focuses on the direct prediction of the aesthetic score distribution. Cumulative distribution function with Jensen–Shannon divergence (CJS) loss was proposed to boost the performance of models [33]. Talebi et al. [10] regarded the score distribution as an ordered class. They used squared EMD loss to predict the score distribution. In this paper, we minimize EMD loss to make the results more accurate.

## 3. Proposed Method

Figure 1 shows the overall framework of the FF-VEN proposed in this paper. The VE module uses the excitation support method to extract ROI from images, aiming to obtain top-down neural attention of ResNet50. Based on ROI, the adaptive filter selects either a Laplace filter or Gaussian filter. It also adjusts the parameters of filters depending on

the degree of visual interest. SDFF module dynamically fuses the shallow feature and the deep feature of VGG16 [12], extracted via transverse connection. In FFU, the pooling layers are used for the corresponding features. FFU calculates the weights of contribution to information by analyzing the statistical characteristics of the features. Next, the pooled features are dotted with their contribution weights, and then put into fusion. Finally, EMD loss is selected to predict the score distribution.

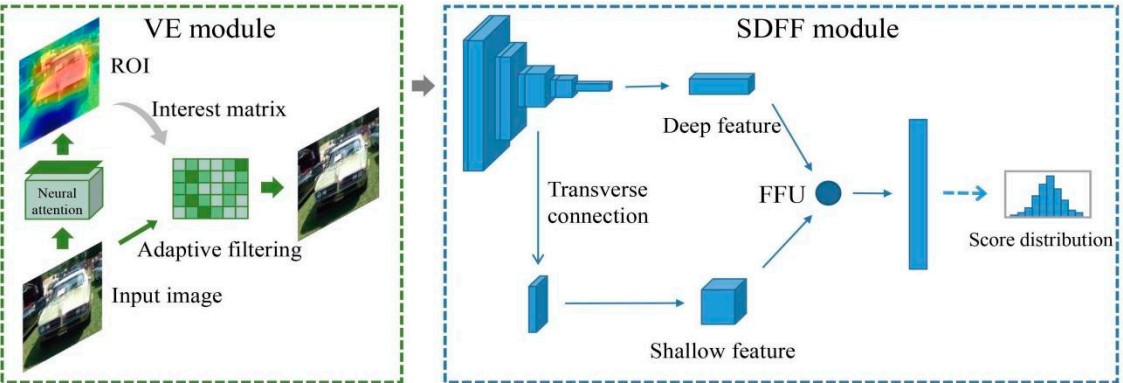

**Figure 1.** The overall framework of FF-VEN. VE module filters images adaptively based on ROI. SDFF module uses FFU to fuse the shallow feature and the deep feature extracted by the method of transverse connection. Finally, the score distribution is directly predicted.

### 3.1. Top-Down Neural Attention

In this paper, ROI represents the level of interest of all pixels in the form of a two-dimensional matrix. The interest degree is calculated via top-down neural feedback from the decisive pixels to all pixels of the original image. For the computer, the interest matrix shows the region with the prominent feature that the CNN pays attention to when predicting. For humans, the value of the interest matrix represents the degree of attraction to the pixel by human eyes. On the basis of the probabilistic winner-take-all (WTA) model, the incentive support method [30] can calculate the interest matrix with the same size as the input image. In statistical concepts, the marginal winning probability $P(o_i)$ represents the attention rate transmitted from the decisive pixels, i.e.,

$$P(o_i) = \sum_{j=1}^{N} P(o_i|o_j) \tag{1}$$

where $o_i$ is a pixel of the overall pixel set in the input image, $N$ is the number of the decisive pixels in the upper layer generated by $o_i$. In (1), $P(o_i)$ sums the pixel's effect degree after quantization between the two layers. In the excitation backprop algorithm, neurons transmit signals through the excitation propagation. The marginal winning probability $P(o_i)$ is obtained via the top-down connections based on the conditional winning probability $P(o_i|o_j)$. If the excitation connection $m_{i,j}$ exists, $P(o_i|o_j)$ is defined as:

$$P(o_i|o_j) = m_{i,j}\hat{o}_i c_j \tag{2}$$

where $m_{i,j}$ represents the connection weight between $o_i$ and $o_j$, $\hat{o}_i$ means the response of $o_i$, and $c_j$ is the normalization factor. According to (1) and (2), the recursive propagation of top-down signals can calculate the interest matrix of images layer by layer. The interest matrix represents ROI in pixels when the CNN makes decisions. In this paper, pre-trained ResNet50 is used to extract the interest matrix. Some examples are shown in Figure 2. ROI of images is highlighted by the pseudo-color technique. In Figure 2, ROI not only distinguishes between the foreground and the background, but also displays the degree of neural attention.

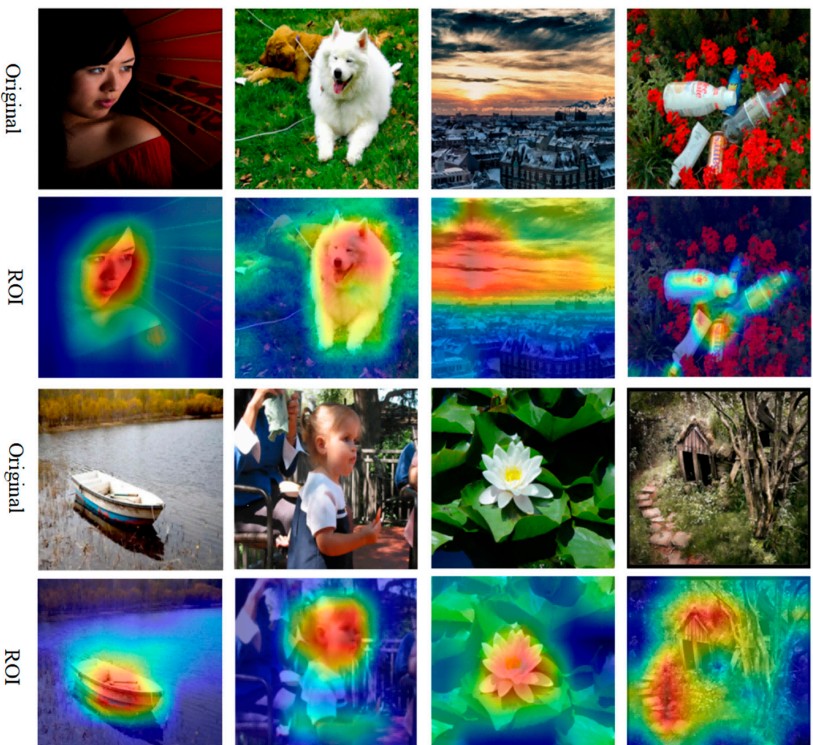

**Figure 2.** Examples of images with ROI. We apply pseudo-color technique to the interest matrix (JET mapping). The colors are red, orange, yellow, green, blue, and purple in turn. Red indicates the highest degree of interest and purple represents the lowest degree of interest.

### 3.2. Adaptive Filtering

An adaptive filter is designed to simulate human eyes based on the fovea visual characteristics. The spatial domain filters conform to the convolution process of the CNN, so the adaptive filter uses Laplace and Gaussian filters. The outermost edge of the image is retained to keep the image size unchanged. As shown in Figure 3, adaptive filtering is carried out on the basis of the interest matrix extracted in Section 3.1. First, the interest matrix is processed via the min-max normalization method. The value of the threshold is set as the average of the interest matrix. Experiments show that the average value accounts for about 60% of the maximum value. Next, the interest degree of each pixel is compared to the value of the threshold, selecting to sharpen or to blur.

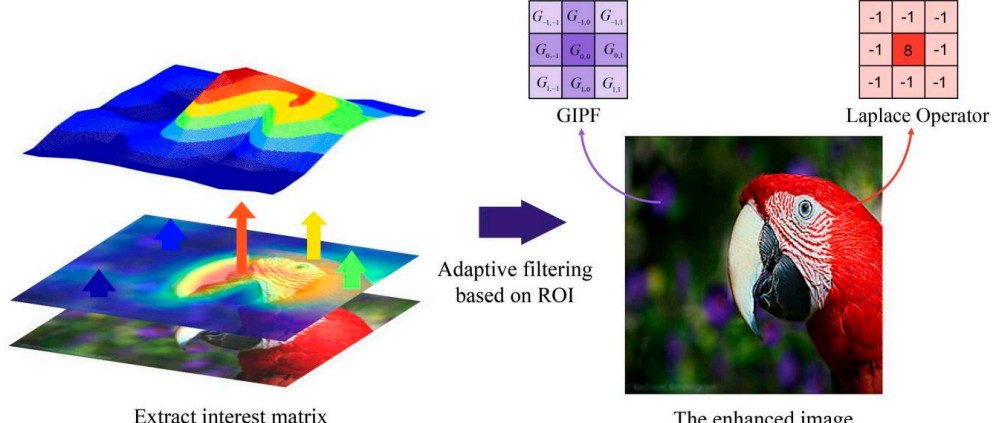

**Figure 3.** The process of adaptive filtering. The adaptive filter analyzes the interest degree in each pixel depending on the extracted interest matrix. GIPF or Laplace filter is dynamically selected to simulate human eyes.

On the one hand, for the process of sharpening, the Laplace operator is used to calculate the details of the images. The Laplace operator of 4 neighborhood pixels is defined as:

$$g_{L4}(x,y) = f(x,y-1) + f(x,y+1) + f(x-1,y) + f(x+1,y) - 4f(x,y) \tag{3}$$

where $f(x,y)$ is the pixel located at coordinates $(x,y)$. There is another kind of expression of a Laplace operator. Its definition is shown below:

$$g_{L8}(x,y) = \sum_{i=-1}^{1} \sum_{j=-1}^{1} f(x+i,y+j) - 9f(x,y) \tag{4}$$

where $g_{L8}(x,y)$ means the Laplace operator with diagonal distribution. $g_{L8}(x,y)$ detects more details and texture, combining fine-grained attributes of 8 neighborhood pixels. In addition, irregular noise belongs to fine-grained information in the spatial domain. Due to the noise's impact on image assessment, the high-pass filter processes images directly. The high-boost filtering combines the original images and the weighted results of Laplace filtering. It linearly enhances the texture and the details of images, i.e.,

$$g(x,y,k,b) = f(x,y) + k \cdot g_L(x,y) + b \tag{5}$$

where $g_L(x,y)$ represents the pixel after Laplace filtering and $f(x,y)$ means a pixel of the input images. $b$ and $k$ are coefficients of the high-boost filtering, and their values depend on the degree of the neural attention. In (5), the high-boost filtering adds the fine-grained texture (obtained by the Laplace operator) to the original pixel. The greater interest degree causes the greater enhancement of texture and details. On the other hand, a two-dimensional Gaussian low-pass filter (GIPF) is utilized for the blurring process:

$$G(x,y,\sigma) = \frac{1}{2\pi\sigma^2} e^{\frac{(x^2+y^2)}{2\sigma^2}} \tag{6}$$

where $x$ and $y$ are the coordinates of pixels, $\sigma$ is the standard deviation of GIPF and its value is determined by the interest matrix. According to (6), the smaller the value of $\sigma$, the severer the peak's change in Gaussian function, and the lower the degree of blurring. On the contrary, a larger value of $\sigma$ results a the higher degree of blurring. Table 1 shows the specific parameters of the filters. *Max* is the highest interest degree of the input image and *threshold* is set to choose the corresponding filter.

**Table 1.** The parameters in the adaptive filter.

| Filter | Size | $k$ | $b$ | $\sigma$ |
|---|---|---|---|---|
| The high-boost filter (including Laplace filter) | $9 \times 9$ | $\frac{i-threshold}{Max-threshold}$ | $2 \cdot (i - threshold)$ | - |
| Gaussian filter | $9 \times 9$ | - | - | $\frac{threshold-i}{threshold-Min}$ |

As mentioned above, the adaptive filter in the spatial domain with contrast processing achieves the goal of visual enhancement. Figure 4 shows some examples of this step. Column 2 shows the quadrupling of the results. In Figure 4, the adaptive filter sharpens or blurs the images to different degrees based on ROI. The process of sharpening leads to a brighter foreground and sharper details. The result of blurring is weakening the presence of the background. For computer vision, the adaptive filter increases the difference between pixels of different interest levels based on the feedback after identifying the object. The cooperation of neural attention in Section 3.1 and the adaptive filtering in Section 3.2 takes advantage of the underlying physiological responses that drive behavior in the human consciousness.

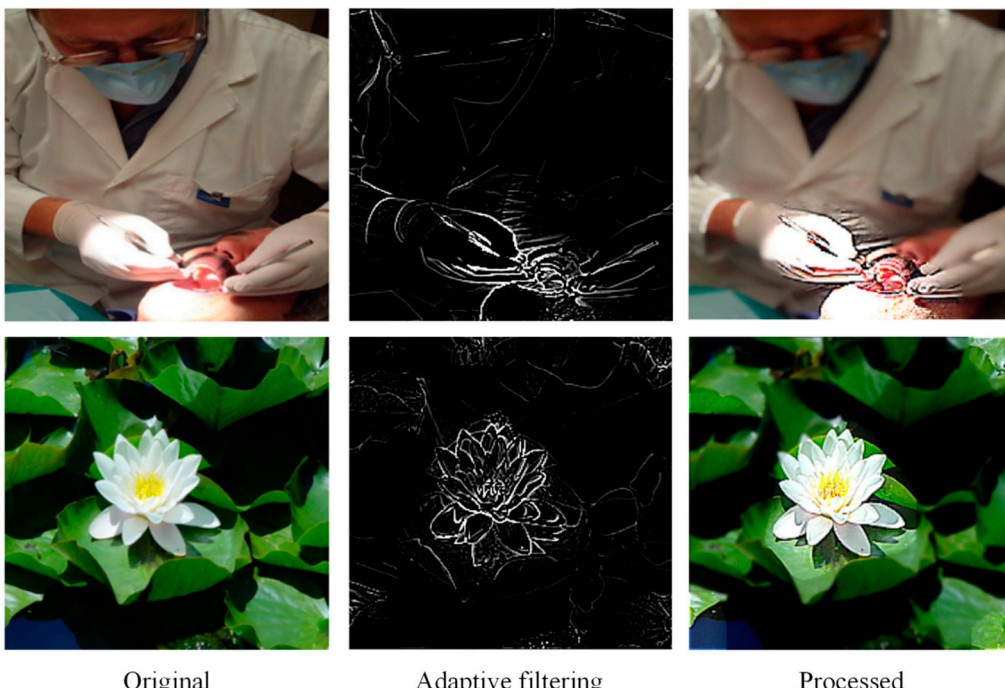

Original  Adaptive filtering  Processed

**Figure 4.** Examples of adaptive filtering. Column 1, the original images; Column 2, the results of adaptive filtering; Column 3, the processed images.

### 3.3. Features at Different Stages

Current studies have found that there are different meanings of features learned by a CNN at different learning stages [13]. The shallow feature contains the low-level information of images, such as color, edge, and texture. The deep feature perceives abstract semantic information. Zhang et al. fuse an SPP layer and sobel-based attention layer. The output of the SPP layer is the result of down sampling the input image [16]. Moreover, they fuse the output of the feed-forward peripheral subnet and the output of the foveal subnet [17]. The output of the feed-forward peripheral subnet is low-resolution, and it goes through several convolution layers. The output of the SPP layer and the output of the feed-forward peripheral subnet are shallow features with a few convolution layers [16,17]. Similar to InceptionNet [9], the SDFF module broadens the network structure, aiming to improve the performance of models. In CNNs, VGGNet is a neural network stacked with convolutional layers. VGGNet has a relatively simple network structure. We use VGG16 [12] as the baseline and extract the shallow feature and the deep feature from different convolution layers. Figure 5 shows an example of the results. The main parameters of VGG16 are listed in Table 2. The max pooling layer after each convolution layer is omitted. From the Conv3-256* layer, we extract the shallow feature, whose size is $28 \times 28 \times 256$ after passing through the max pooling layer. The deep feature is taken from the Conv3-512* layer. Its size is $7 \times 7 \times 512$. The above process is mathematically expressed as:

$$\begin{cases} I_1' = Pl_1(\psi(W_1 \cdot I_0)) \\ I_2' = Pl_2(\psi(W_2 \cdot I_1)) \end{cases} \quad (7)$$

where $I_0$ represents an input image, $I_1'$ and $I_2'$ are the output of the transverse connection, $\psi(W_i \cdot I_{i-1})$ is the state function of VGG16, and $Pl_i(I_{i-1})$ is the feature pooling function with $i = 0, 1, 2$. In (7), $I_1$ is taken out when VGG16 is the state function $\psi(W_1 \cdot I_0)$. Via the feature pooling function $Pl_1(I_1)$, the shallow feature $I_1'$ is obtained. Similarly, the deep feature $I_2'$ is extracted using the method of transverse connection. Adding the shallow feature reduces the influence of the deep feature on the results. In this way, the low-level and semantic information of the images can be integrated to improve the network performance.

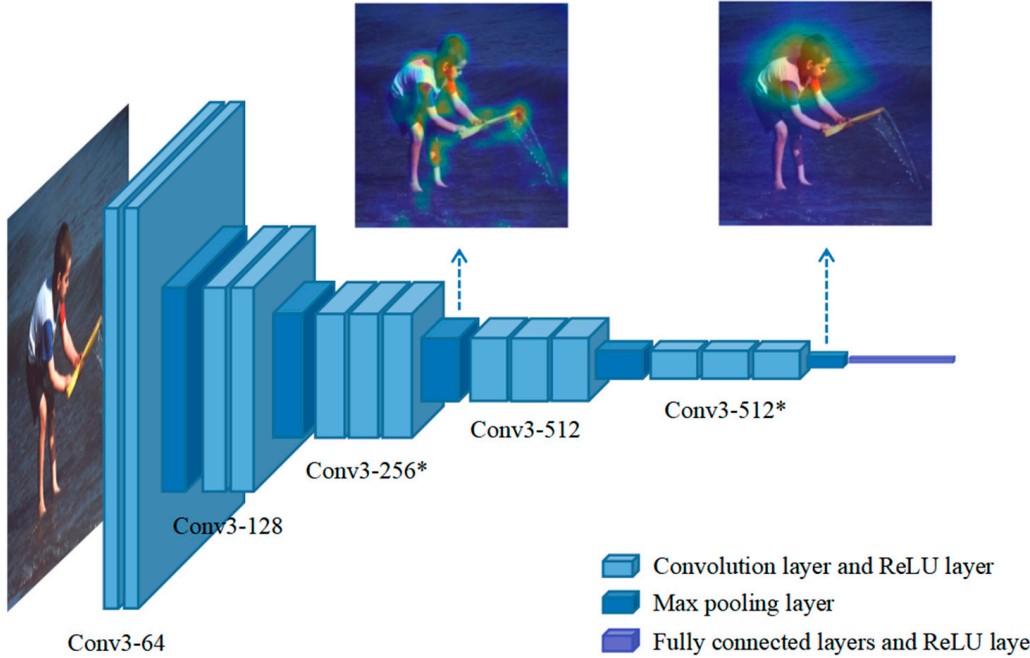

**Figure 5.** The feature's understanding of an image in VGG16. The information contained in the features is attached to the original image with the incentive support method. It is confident for the deep feature to recognize objects. The shallow feature captures the foreground by learning low-level information. The asterisk * represents the layer that we take out the feature from.

**Table 2.** The main parameters of VGG16.

| Layer [a] | The Size of Input Data | The Number of the Layer |
|---|---|---|
| Conv3-64 | $224 \times 224 \times 3$ | 2 |
| Conv3-128 | $112 \times 112 \times 64$ | 2 |
| Conv3-256 * | $56 \times 56 \times 128$ | 3 |
| Conv3-512 | $28 \times 28 \times 256$ | 3 |
| Conv3-512 * | $14 \times 14 \times 512$ | 3 |

[a] The layers from Line 1 to Line 5 are the convolution layer of VGG16. The asterisk * represents the layer that we take out the feature from.

### 3.4. Feature Fusion Unit

After extracting the features, SDFF module needs a feature fusion mechanism to combine the shallow feature and the deep feature. Figure 6 shows FFU after fine-tuning. PCFS means a pooling layer, a catenation layer, a fully connected layer, and a sigmoid layer in turn. The pooling layer analyzes the statistical characteristics of the features. The next layers calculate the weights (denoted by $k_x$ with $x = s$ or $d$ in Figure 6) of the pooled features. $s$ means the shallow feature and $d$ is the deep feature. $k_x$ represents the contribution weight of the feature to the information. Then, the shallow feature and the deep feature pass through the max pooling layer and the average pooling layer, respectively. Max pooling not only selects the data with higher recognition but also provides the nonlinearity factor for FFU. The deep feature is the result that the CNN learns in the later stage, so it influences the CNN greatly. Average pooling considers all of the deep information. Because the sizes of the two pooling layers are $7 \times 7$, the shallow feature and the deep feature are rescaled to $7 \times 7 \times 256$ and $7 \times 7 \times 512$. Afterwards, we take the dot product of each pooled feature and its $k_x$. Finally, the results are fused by the catenation layer. The main parameters of FFU are showed in Table 3.

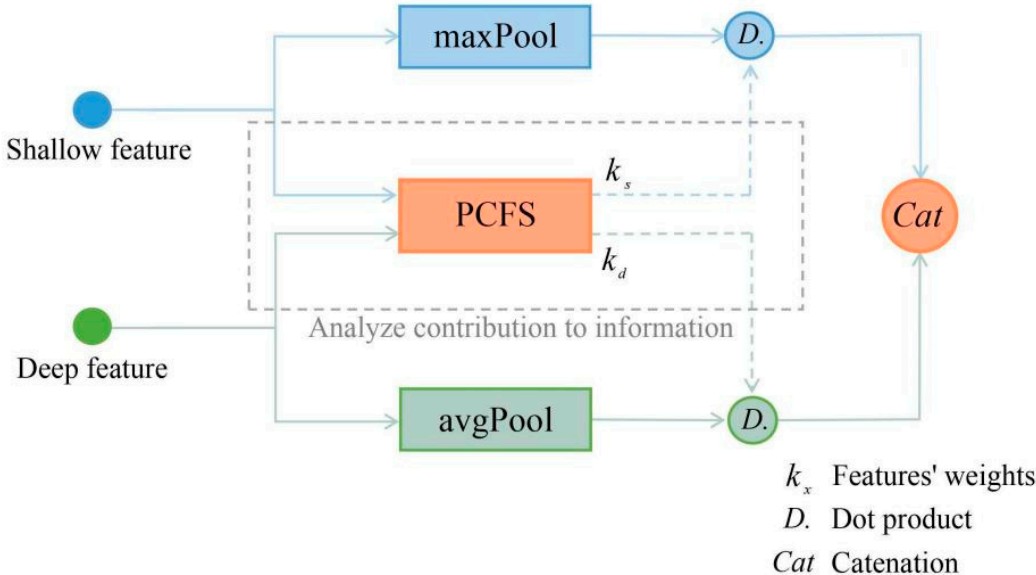

**Figure 6.** The framework of FFU. PCFS analyzes the contribution of features. Meanwhile, features pass through the max pooling layer and the average pooling layer, respectively. FFU dot pooled features with their weights and then fuse the results via the catenation layer. The gray dotted box represents the process of analyzing the contribution without changing the features numerically.

**Table 3.** The main parameters of FFU.

| Layer | The Size of Input Data | The Number of the Layer |
|---|---|---|
| PCFS | $28 \times 28 \times 256, 7 \times 7 \times 512$ | 1 |
| FC | $7 \times 7 \times 256 + 7 \times 7 \times 512$ | 1 |

### 3.5. EMD Loss

In the AVA dataset and Photo.net dataset, the score distribution is intrinsically ordered. For ordered classes, the performance of classification models is better than regression frameworks. However, the classification task ignores the relationships between classes of score distribution. EMD loss penalizes misclassifications according to class distances. In this paper, EMD loss is minimized to predict the score distribution directly. Because of the impact of the number of assessors on credibility, the distribution is normalized. The definition of EMD loss is shown below:

$$\text{EMD}(l, p) = \left( \frac{1}{N} \sum_{i=1}^{N} \left| CDF_l(i) - CDF_p(i) \right|^r \right)^{\frac{1}{r}} \tag{8}$$

where $CDF_x(i)$ represents the cumulative distribution function as $\sum_{n=1}^{i} e_{d_n} (1 \leq i \leq N)$. $d_n$ means the $n$th normalized number of assessors. $x = l, p$ ($l$ is the label distribution and $p$ is the predicted distribution). In (8), EMD is the minimum distance between the mass of two score distributions. We set $r$ as 2 to punish the Euclidean distance between $CDFs$, aiming to optimize the network.

### 4. Experiments

In this section, the performance of FF-VEN is evaluated on AVA dataset and Photo.net dataset. Compared with previous studies, FF-VEN is a promising model for IAA.

### 4.1. Datasets

AVA dataset [34]: the AVA dataset is a popular dataset for IAA because of the large number of images, the diversity of content, and the consistency of data. It can be seen at http://www.dpchallenge.com/ (accessed on 11 December 2021). For an image, there are 66 semantic labels, 14 style labels, and a label distribution with 10 scores (from 1 to 10). In the AVA dataset, higher scores mean higher quality. For an image with the average score in a certain interval, its score distribution tends to be Gamma or Gaussian [34]. Figure 7 shows some examples from the AVA dataset. On average, each image is assessed by about 200 people, including professional image workers, photographers, and photography enthusiasts. The AVA dataset contains more than 250,000 images. We removed the images whose variance is high or whose average score is 5. Thus, 235,086 images were used for training, 18,987 for verification, and 1000 for testing.

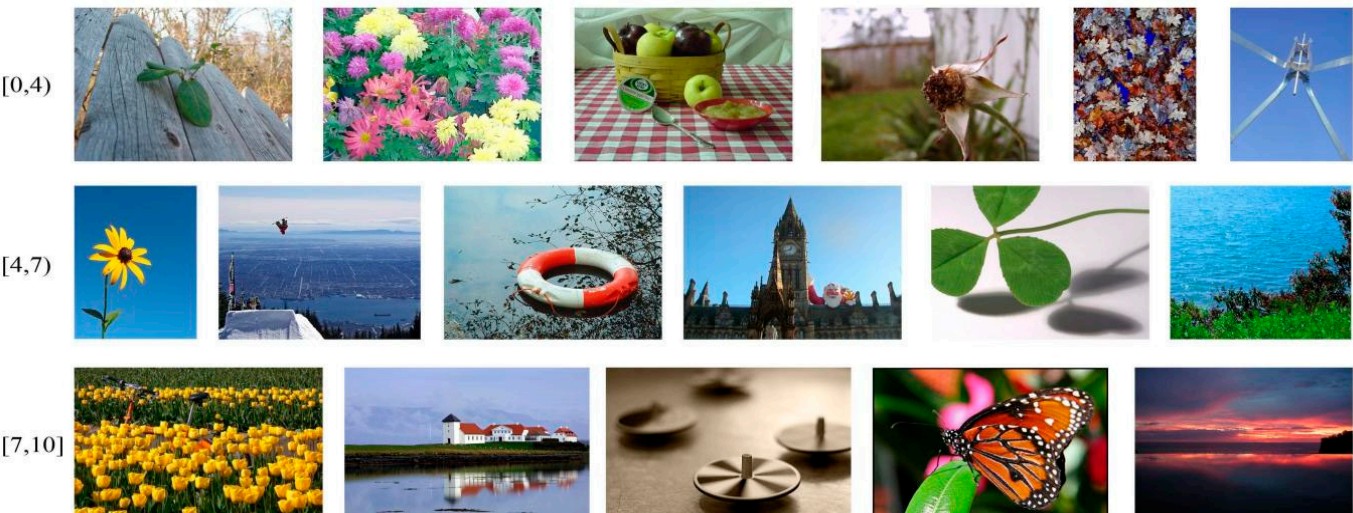

**Figure 7.** Examples of images with the average score in different intervals in AVA dataset. Line 1, the images with the average score in [0, 4); Line 2, the images with the average score in [4, 7); Line3, the images with the average score in [7, 10].

Photo.net dataset [35]: the Photo.net dataset contains about 20,000 images. We collected them from https://www.photo.net/ (accessed on 11 December 2021), a platform for photography enthusiasts to share images. This website offers discussion forums, image reviews, galleries, etc. People assess images based on aesthetics and creativity, with a score between 1 and 7 for each. Photo.net explains that 1 means low quality and 7 means high quality. Reasons for a high score include rich colors, interesting composition, and eye-catching content. In the Photo.net dataset, the images are diverse, which is a challenge for deep learning. Excluding invalid images and lost images, 16,663 images were obtained by crawlers. 14,000 images were used for training, 1000 for verification, and the remaining 1663 images were used for testing.

### 4.2. Details of the Experiment

The size of the input images is $224 \times 224 \times 3$. The images are resampled by the ANTIALIAS algorithm of PIL package in PYTHON library. Batch size is 16, initial learning rate is $1 \times 10^{-3}$, momentum is 0.9, learning decay rate is 0.0002, and epoch is 10. The number of iterations of the AVA dataset is 14,693 and that of the Photo.net dataset is 1042. Our network is based on the open source TorhchRay, Caffe, and PyTorch frameworks. We use a single NVIDIA GeForce GTX 1650 GPU.

Based on the direct prediction of the score distribution, we evaluate FF-VEN in the classification task and the regression task. In the regression task, we use these indicators: Pearson linear correlation coefficient (LCC), Spearman rank-order correlation coefficient

(SRCC), mean absolute error (MAE), and root mean square error (RMSE). The evaluation index formulas are shown as:

$$\begin{cases} \text{LCC} = \frac{1}{N-1} \sum_{i=1}^{N} \left( \frac{l_i - \bar{l}}{\sigma_l} \right) \left( \frac{p_i - \bar{p}}{\sigma_p} \right) \\ \text{SRCC} = \frac{\sum_{i=1}^{N} (l_i - \bar{l})(p_i - \bar{p})}{\sqrt{\sum_{i=1}^{N} (l_i - \bar{l})^2 \sum_{i=1}^{N} (p_i - \bar{p})^2}} \\ \text{MAE} = \frac{\sum_{i=1}^{N} |p_i - l_i|}{N} \\ \text{RMSE} = \sqrt{\frac{\sum_{i=1}^{N} (p_i - l_i)^2}{N}} \end{cases} \tag{9}$$

where $l$ is the label distribution and $p$ is the predicted distribution, $\bar{l}$ is the average of $l$, $\sigma_l$ is the standard deviation of $l$, $\bar{p}$ is the average of $p$, $\sigma_p$ is the standard deviation of $p$. LCC applies to normally distributed data to predict the accuracy of the model. SRCC is suitable for nonlinear data. It calculates the correlation of the scores in the corresponding position in arrays between the prediction distribution and the label distribution. The values of SRCC and LCC vary between 0 and 1. The larger value means the better model performance. RMSE measures the deviation between the predicted results and the labels. MAE calculates the average of residuals directly. MAE and RMSE are expected to be smaller. In the classification task, we calculate *Mean* of the score distribution and compare it with the value of the threshold. We define *Mean* as:

$$Mean = \sum_{i=1}^{N} s_i \times i \tag{10}$$

where $s_i$ is the score when the class of the distribution is $i$. $N$ is 10 when AVA dataset and 7 when Photo.net dataset. The value of the threshold is set as 5, as Ma et al. did in [15]. Images with the value of *Mean* above 5 are regarded as high quality. Otherwise, they are classified as low quality images. In the classification task, the selected index is *Accuracy*, i.e.,

$$Accuracy = \frac{TP + TN}{P + N} \tag{11}$$

where $P$ is positive cases, $N$ is negative cases, $TP$ is true and positive cases, and $TN$ is true and negative cases.

### 4.3. Comparison on AVA Dataset

We compare FF-NET with other models on AVA dataset. The results are shown in Table 4. SPP-Net is a network with spatial pyramid pooling for the pretreatment of images [36]. AA-Net is a cropping model with attention box prediction (ABP) [37]. Zhang et al. [17] recorded the evaluation results of SPP-NET based on VGG16 [12]. In the classification task, the accuracy of FF-VEN is 83.64%, 9.23% higher than that of SPP-Net, 6.64% higher than that of AA-Net. Compared with SPP-Net, the LCC of FF-VEN is 31.7% larger, the SRCC is 25.7% larger, and the EMD is 23.9% smaller. The MAE and RMSE are slightly better than SPP-Net and AA-Net. The contrast between them suggests the superiority of our network. We list three advanced methods: NIMA [10], ResNet [37], and InceptionNet [9]. In their experiments, the network on the basis of InceptionNet performed best, with an accuracy more than 2% greater than InceptionNet. Specifically, NIMA outperforms InceptionNet by 2.08%, demonstrating that it is helpful to broaden the network structure of CNN. The LCC and SRCC of GPF-CNN [17] are 2.6% higher and 2.1% higher, which reveals that neural attention benefits the computer in assessing images from the perspective of human eyes. For ReLIC++ [29], accuracy, LCC and SRCC are 82.35%, 0.76 and 0.748, respectively. This indicates the advantages of FFU. In addition, ReLIC++

has a deeper understanding of the features of images. These successful cases verified the rationality of FF-VEN. The accuracy of FF-VEN is 4.21% higher than InceptionNet. Additionally, our network is superior to previous studies in the regression task. This shows the effectiveness of FF-VEN.

**Table 4.** The results of comparison on AVA dataset.

| Network Architecture | Accuracy (%) | LCC | SRCC | MAE | RMSE | EMD |
|---|---|---|---|---|---|---|
| SPP-Net [36] | 74.41 | 0.5869 | 0.6007 | 0.4611 | 0.5878 | 0.0539 |
| AA-Net [37] | 77.00 | - | - | - | - | - |
| InceptionNet [9] | 79.43 | 0.6865 | 0.6756 | 0.4154 | 0.5359 | 0.0466 |
| NIMA [10] | 81.51 | 0.636 | 0.612 | - | - | 0.050 |
| GPF-CNN [17] | 81.81 | 0.7042 | 0.6900 | 0.4072 | 0.5246 | 0.045 |
| ReLIC++ [29] | 82.35 | 0.760 | 0.748 | - | - | - |
| FF-VEN | 83.64 | 0.773 | 0.755 | 0.4011 | 0.5109 | 0.044 |

### 4.4. Comparison on Photo.net Dataset

On the Photo.net dataset, FF-VEN is compared with GIST-SVM [38], FV-SIFT-SVM [38], MRTLCNN [39], and GLFN [16]. The results are shown in Table 5. Marchesotti et al. [38] used the generic image descriptors to assess images and treated IAA as a classification problem. However, the indices of two kinds of SVM are around 60% for the classification task. The accuracy of FF-VEN is 78.1%, which is obviously better than the networks based on SVM. For the deep learning networks (MRTLCNN, GLFN), we all chose VGG16 [12] as the baseline, similar to [16]. MRTLCNN is a multi-task framework that combines aesthetic labels and semantic labels [39]. The accuracy of FF-VEN is 12.9% higher than that of MRTLCNN and 2.5% higher than that of GLFN. In the regression task, LCC is 16.7% better and SRCC is 18.3% better. This indicates that FF-VEN outperforms GLFN on small-scale datasets such as the Photo.net dataset.

**Table 5.** The results of comparison on Photo.net dataset.

| Network Architecture | Accuracy (%) | LCC | SRCC | MAE | RMSE | EMD |
|---|---|---|---|---|---|---|
| GIST-SVM [38] | 59.9 | - | - | - | - | - |
| FV-SIFT-SVM [38] | 60.8 | - | - | - | - | - |
| MRTLCNN [39] | 65.2 | - | - | - | - | - |
| GLFN [16] | 75.6 | 0.5464 | 0.5217 | 0.4242 | 0.5211 | 0.070 |
| FF-VEN | 78.1 | 0.6381 | 0.6175 | 0.4278 | 0.5285 | 0.062 |

### 4.5. Evaluation of Two Sub-Modules

The adaptive filter in 3.2 and feature fusion unit in 3.4 correspond to the VE module and SDFF module, respectively. VE-CNN (VGG16) adds the VE module on the basis of the original VGG16. It means that VE-CNN (VGG16) is the result of FF-VEN without the adaptive filter. SDFF (VGG16) takes out the features in VGG16 and then fuses them. It means that SDFF (VGG16) is the result of FF-VEN without feature fusion unit. We compare two sub-modules with VGG16 [40], Random-VGG16 [22], Saliency-VGG [41], and GPF-CNN (VGG16) [17]. The results on the AVA dataset are shown in Table 6. Saliency-VGG16 combined the global and local information according to the saliency map [40]. In Table 6, Random-VGG16 outperforms VGG16, indicating randomness improves the performance of models. In accuracy, Saliency-VGG16 is 79.19% and GPF-CNN is 80.70%. This shows the importance of neural attention. VE-CNN (VGG16) is superior to previous studies in the regression task. LCC is 7.5% higher than GPF-CNN (VGG16) and SRCC is 6.25% higher. This suggests that adaptive filtering based on ROI helps FF-VEN to process the details of images. We use ResNet50 to extract ROI from images. The neural attention of

ResNet50 benefits the network performance of VGG16. SDFF (VGG16) performs slightly better than GPF-CNN (VGG16) in the classification task. Additionally, accuracy is 7.06% better. However, it does not perform as well as VE-CNN for LCC and others. This indicates that SDFF module broadens the network structure of VGG16, deepening the memory of FF-VEN and reducing the number of required samples.

**Table 6.** The experimental results of sub-modules on AVA dataset.

| Network Architecture | Accuracy (%) | LCC | SRCC | MAE | RMSE | EMD |
|---|---|---|---|---|---|---|
| VGG16 [40] | 74.41 | 0.5869 | 0.6007 | 0.4611 | 0.5878 | 0.0539 |
| Random-VGG16 [22] | 78.54 | 0.6382 | 0.6274 | 0.4410 | 0.5660 | 0.0510 |
| Saliency-VGG16 [40] | 79.19 | 0.6711 | 0.6601 | 0.4228 | 0.5430 | 0.0475 |
| GPF-VGG16 [17] | 80.70 | 0.6868 | 0.6762 | 0.4144 | 0.5347 | 0.0460 |
| VE-CNN (VGG16) | 81.03 | 0.7395 | 0.7185 | 0.4073 | 0.5279 | 0.0441 |
| SDFF (VGG16) | 81.47 | 0.7119 | 0.7021 | 0.4103 | 0.5317 | 0.0462 |

*4.6. Quality-Based Comparison*

As mentioned in [34], the score distribution of images with *Mean* in the range of [0, 4) or [7, 10] tends to be Gamma. The number of those images account for 4.5% of all images. If *Mean* in the range [4, 7), the score distribution of the corresponding image is largely Gaussian. Inspired by this, we divide AVA dataset into three parts depending on *Mean* and conduct the experiments respectively. The results are shown in Table 7. In the interest of fairness, we adopt VGG16 as the basic model. For *Mean* of [4, 7), MAE of FF-VEN is 0.3748 and LCC is 0.8945. It indicates that the larger the number of images, the more consistent the scores predicted by CNN with the labels. As the score distribution of most images is Gaussian, the prediction of CNN tends to be Gaussian. As a result, the performance of CNN is poor in assessing images with Gamma distribution. It is worth noting that the accuracy of the three models is greater than 90% for images with *Mean* in [7, 10]. Because professional images are excellent in composition, tone, and other aspects, CNN is more likely to distinguish them. The accuracy of FF-VEN is 3.78% higher than NIMA [10] and 2.07% higher than ReLIC++ [29]. For professional images, this suggests that FF-VEN captures the object's contour and increases the gap between the foreground and the background effectively.

**Table 7.** Evaluation results for images with *Mean* in different intervals.

| *Mean* | Network Architecture | Accuracy (%) | LCC | SRCC | MAE | RMSE | EMD |
|---|---|---|---|---|---|---|---|
| | NIMA [10] | 78.46 | 0.6265 | 0.6043 | 0.5577 | 0.6897 | 0.067 |
| [0, 4) | ReLIC++ [29] | 80.02 | 0.6887 | 0.6765 | - | - | - |
| | FF-VEN | 80.59 | 0.7095 | 0.6971 | 0.5037 | 0.6139 | 0.059 |
| | NIMA [10] | 80.43 | 0.7271 | 0.7028 | 0.4037 | 0.5256 | 0.048 |
| [4, 7) | ReLIC++ [29] | 81.15 | 0.8733 | 0.8547 | - | - | - |
| | FF-VEN | 81.33 | 0.8945 | 0.8831 | 0.3748 | 0.4851 | 0.039 |
| | NIMA [10] | 94.93 | 0.5936 | 0.5645 | 0.5927 | 0.7314 | 0.073 |
| [7, 10] | ReLIC++ [29] | 96.64 | 0.6223 | 0.6084 | - | - | - |
| | FF-VEN | 98.71 | 0.6113 | 0.6492 | 0.5343 | 0.6457 | 0.061 |

Figure 8 shows some examples of images with *Mean* in different intervals for comparison. It can be seen that the difference between the distribution predicted by FF-VEN and that of the labels is smaller than the other two. Images with *Mean* in [7, 10] are less controversial. Most people give these images high scores. The composition of professional images is abstract and artistic, which is difficult for CNN to learn. From the above experiments, it seems that the VE module enhances the features of images based on human

visual characteristics, leading to improving the prediction confidence of FF-VEN. Figure 9 shows some failure cases of FF-VEN. The network we trained does not perform well on images with very non-Gaussian distributions, such as bimodal or very skewed distributions. However, the Gaussian functions perform adequately for 99.77% of all the images in the AVA dataset [17].

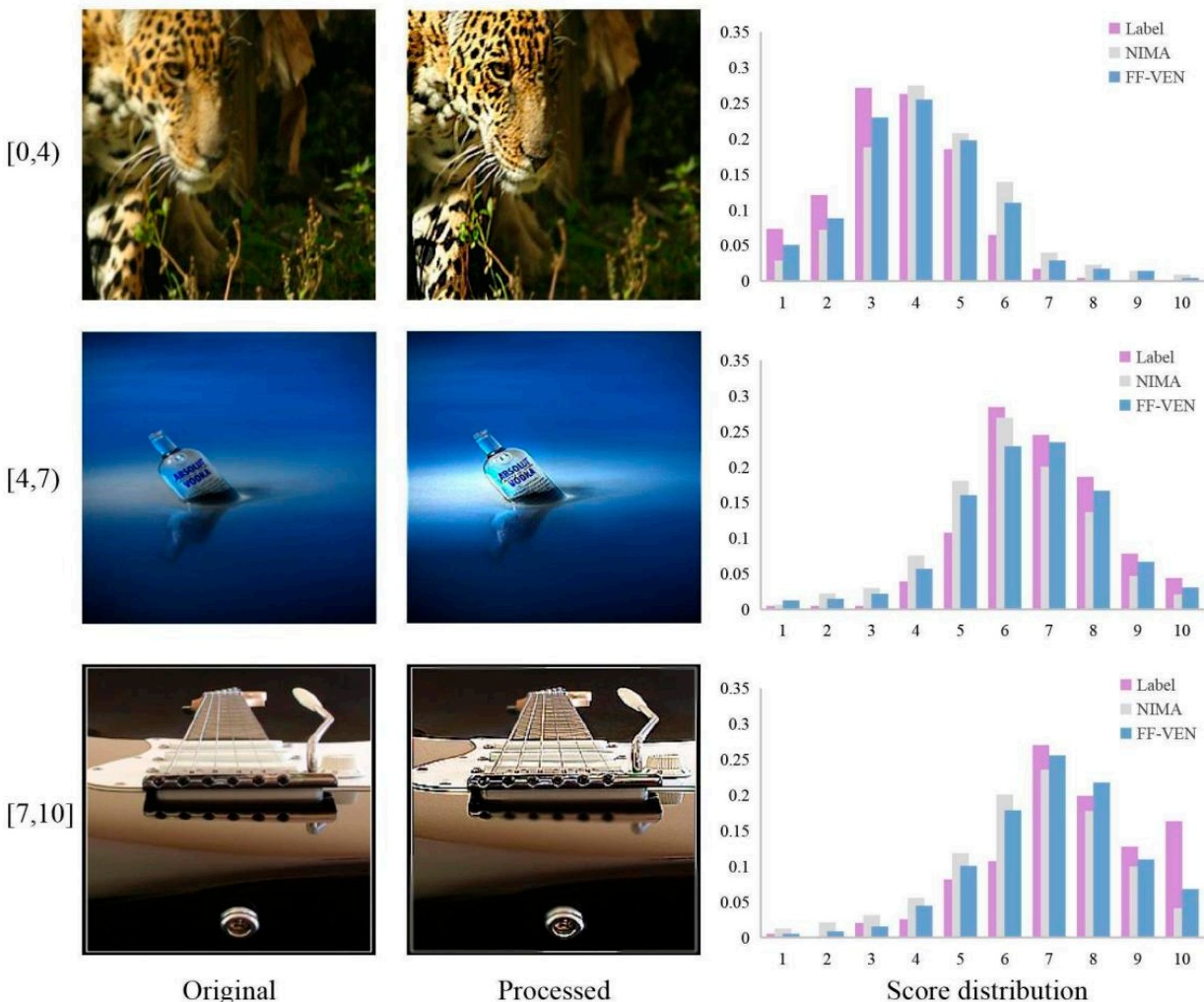

**Figure 8.** Some examples of the results of FF-VEN. *Mean*s of the images in Line 1, Line 2, and Line 3 are, respectively in [0, 4), [4, 7), [7, 10]. In Column 3, the magenta scores are the label distribution. The gray distribution is predicted by NIMA and the mazarine distribution is predicted by FF-VEN.

### 4.7. Different Shallow Features

We conducted ablation studies on other layers in VGG16. According to the network characteristics of VGG16 in Table 2, the deep feature is the final output of VGG16, that is, without considering the layer Conv3-512 which input size is $14 \times 14 \times 512$. For the other four network layers, the output data are assumed to be shallow features. We keep the other network structures of FF-VEN unchanged and fuse these shallow features and the deep feature separately. The experimental results on AVA dataset are shown in Table 8. It shows that the output of the two specific layers, Conv3-256 with input size $56 \times 56 \times 128$ and Conv3-512 with input size $14 \times 14 \times 512$, can give the FF-VEN the optimal performance.

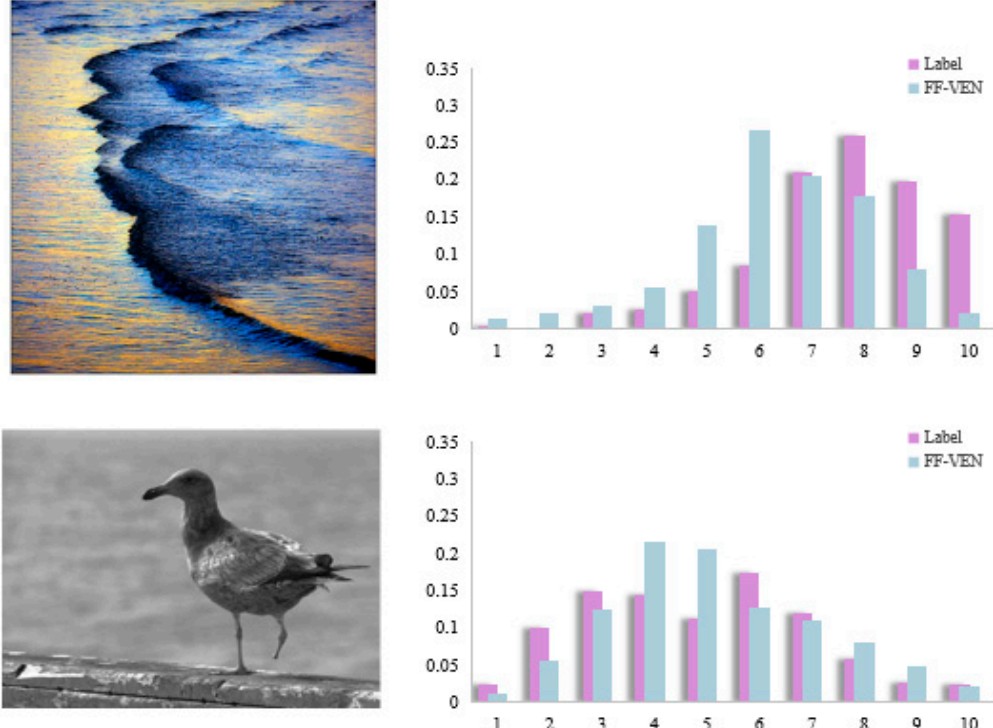

**Figure 9.** Some examples of failure cases.

**Table 8.** Experiments in the shallow features in other layers of VGG16.

| Layer | Accuracy (%) | LCC | SRCC | MAE | RMSE | EMD |
|---|---|---|---|---|---|---|
| Conv3-64 | 80.21 | 0.692 | 0.682 | 0.4163 | 0.5376 | 0.046 |
| Conv3-128 | 81.47 | 0.716 | 0.691 | 0.4025 | 0.5284 | 0.045 |
| Conv3-256 | 83.64 | 0.773 | 0.755 | 0.4011 | 0.5109 | 0.044 |
| Conv3-512 | 82.34 | 0.751 | 0.737 | 0.4047 | 0.5201 | 0.044 |

*4.8. Model Size Comparison*

Timings of one pass of NIMA (VGG16) [10] models on an image of size $224 \times 224 \times 3$ are 150.34 ms (CPU) and 85.76 ms (GPU). Additionally, it has 134.3 million parameters. In ReLIC++ [29], the attention map is of size $49 \times 49$. The training time cost of Full GoogLeNetV1-BN [23] is 16 days. The model size is 82.56 million. Training ILGNet-Inc.V1-BN [25] costs 4 days. In manuscripts, the model size of FF-VEN is 14.7 Million. Evidently, FF-VEN is significantly lighter than ReLIC++. The SDFF module improves the model size by about 119.6 M, compared to NIMA (VGG16). Training FF-VEN costs 4 days, which is faster than Full GoogLeNetV1-BN. In general, FF-VEN is light-weight and achieves inspiring aesthetic prediction accuracy, as reported in Table 9.

**Table 9.** Comparison of model size.

| Model | Size |
|---|---|
| NIMA (VGG16) [10] | 134.3 M |
| GoogLeNet [23] | 82.36 M |
| ReLIC++ [29] | 17.51 M |
| FF-VEN | 14.7 M |

## 5. Conclusions

FF-VEN proposed in this paper considers neural attention, human visual characteristics, and image understanding. It consists of a VE module and SDFF module. According to ROI extracted by neural feedback, the VE module not only selects the Laplace filter or GLPF but also adjusts the parameters of filters. It enables the computer to simulate human eyes when assessing digital images. The SDFF module takes out the shallow feature and the deep feature via transverse connection and fuses them on the basis of information contribution maximization. The results of comparison on the AVA dataset and Photo.net dataset demonstrate the superiority of FF-VEN. In the future, we aim to analyze the network structure of ResNet, InceptionNet, and other CNN. To make the method more comprehensive, we intend to focus on more factors, such as image themes, photography aesthetics, and human emotions.

**Author Contributions:** Methodology, X.Z.; Supervision, X.Z.; Validation, X.J.; Writing—original draft, X.J.; Formal analysis, Q.S.; Writing—review and editing, P.Z. All authors have read and agreed to the published version of the manuscript.

**Funding:** This work was supported by National Key Research and Development Program of China (No. 2020AAA0108700).

**Data Availability Statement:** We used publicly available datasets in order to illustrate and test our network. The AVA dataset can be found in http://www.dpchallenge.com/ (accessed on 11 December 2021) and Photo.net dataset can be found in https://www.photo.net/ (accessed on 11 December 2021).

**Conflicts of Interest:** The authors declare that they have no conflict of interest.

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
