# Peer review of "A Visual Enhancement Network with Feature Fusion for Image Aesthetic Assessment"

_electronics, doi:10.3390/electronics12112526_

Round 1
Reviewer 1 Report
This paper proposes a visual enhancement network called FF-VEN for image aesthetic assessment, which is based on feature fusion. The proposed network consists of a visual enhancement module, as well as a shallow and deep feature fusion module. Experimental results show the effectiveness of the proposed method. I have some comments as follows:
1. Since the backbone network has many layers, why choose the outputs from two specific layers for the so-called shallow and deep features? And how about using other layers for feature fusion?
2. The ablation study would be helpful to validate each proposed technique, such as the modules, adaptive filter, and feature fusion strategy, etc.
3. Some numbers are missing in Tables 4-5 and 7.
4. Except for image aesthetic assessment, several recent works on natural image quality assessment IQA are recommended to be reviewed, including GraphIQA, metaIQA, Lifelong IQA, etc.
5. It is suggested to further improve the presentation of this paper. For example, the figures are blurry (such as Figures 8-9). Please proofread the paper and check all figures.
n/a
Author Response
We thank the reviewers for your constructive and insightful comments on our paper and for the opportunity to improve on its quality. We have carefully revised our manuscript in accordance with your comments and suggestions. The answers and explanations of the changes made to the original manuscript are attached.

Reviewer 2 Report
This paper is well written and I am not surprised that the method works with adaptive filters. It would be better if the optimized parameter is suggested for the filters.
Author Response

(The authors gave the same response as above.)

Round 2
Reviewer 1 Report
Thanks for the response. The authors tried to address my comments, but there are still some issues.
1. The definition of shallow and deep features are confusing. Why the output data are assumed to be shallow features for the other layers? Do you have any references about this assumption?
2. The ablation study means removing the specific component while remains other parts and then testing the performance. It seems the authors fail to address this point.
3. What do you mean by some NNs only regard image aesthetic assessment as the classification and regression tasks? The reasons for the missing numbers are unclear.
4. Yes. The traditional IQA is different from IAA, so it would be better to point out this difference and provide some recent literature as I said in the previous comments. From this clarification, one can better design a model for IAA, otherwise, why not adopt the traditional IQA methods here?
5. The texts in these figures are still blurry.
n/a
Author Response
We thank the reviewer for your constructive and insightful comments on our paper and for the opportunity to improve on its quality. We have carefully revised our manuscript in accordance with your comments and suggestions. The answers and explanations of the changes made to the original manuscript are attached.

Round 3
Reviewer 1 Report
Thanks for your response. But there are still some issues.
The authors claimed that the deep features refer to the final convolutional layer, while others are shallow. However, for relatively more deep CNNs, such as ResNet, obviously not only the final convolutional layer is for deep features.
The results of VE-CNN and SDFF are not significantly different, say the SDFF's ACC is better, but it does not perform as well as VE-CNN for LCC and others.
From the authors' response, it seems the output of IAA tasks is finally a regression score. Does any IAA work only for classification without regression?
The statements of IAA is similar to subjective evaluation in IQA is confusing. What do you mean subjective evaluation in IQA focuses on objectively scoring? The IAA models shall also predict the subjective scores of IAA subjective data.
And most IQA methods do not consider data enhancement is not correct. Many IQA models actually adopt this technique since the subjective datasets are usually small.
I agree that using IQA methods for IAA needs further exploration, but at least the authors can point out the differences and add the references in the paper.
n/a
Author Response

(The authors gave the same response as above.)
